# Fructose-Induced Metabolic Dysfunction Is Dependent on the Baseline Diet, the Length of the Dietary Exposure, and Sex of the Mice

**DOI:** 10.3390/nu17010124

**Published:** 2024-12-31

**Authors:** Taghreed Fadhul, Se-Hyung Park, Heba Ali, Yasir Alsiraj, Jibran A. Wali, Stephen J. Simpson, Samir Softic

**Affiliations:** 1Department of Pharmacology and Nutritional Sciences, University of Kentucky College of Medicine, Lexington, KY 40536, USA; tfa235@uky.edu (T.F.); sh.park@uky.edu (S.-H.P.); yaal223@uky.edu (Y.A.); 2Department of Pediatrics and Gastroenterology, University of Kentucky College of Medicine, Lexington, KY 40536, USA; hhma223@uky.edu; 3Faculty of Science, School of Life and Environmental Sciences, The University of Sydney, Sydney, NSW 2006, Australia

**Keywords:** fructose, ketohexokinase, de novo lipogenesis, insulin resistance, metabolic syndrome, MASLD, sex differences

## Abstract

**Background/Objectives:** High sugar intake, particularly fructose, is implicated in obesity and metabolic complications. On the other hand, fructose from fruits and vegetables has undisputed benefits for metabolic health. This raises a paradoxical question—how the same fructose molecule can be associated with detrimental health effects in some studies and beneficial in others. This study investigates how diet and sex interact with fructose to modulate the metabolic outcomes. **Methods:** Male and female mice were fed different normal chow diets, Boston chow diet (BCD; 23% protein, 22% fat, 55% carbohydrates), Lexington chow diet (LXD; 24% protein, 18% fat, 58% carbohydrates), and low-fat diet (LFD; 20% protein, 10% fat, 70% carbohydrates), supplemented with 30% fructose in water. **Results**: Fructose-supplemented male mice on BCD gained weight and developed glucose intolerance and hepatic steatosis. Conversely, male mice given fructose on LXD did not gain weight, remained glucose-tolerant, and had normal hepatic lipid content. Furthermore, fructose-fed male mice on LFD did not gain weight. However, upon switching to BCD, they gained weight, exhibited worsening liver steatosis, and advanced hepatic insulin resistance. The effects of fructose are sex-dependent. Thus, female mice did not gain weight and remained insulin-sensitive with fructose supplementation on BCD, despite developing hepatic steatosis. These differences in metabolic outcomes correlate with the propensity of the baseline diet to suppress hepatic ketohexokinase expression and the de novo lipogenesis pathway. This is likely driven by the dietary fat-to-carbohydrate ratio. **Conclusions:** Metabolic dysfunction attributed to fructose intake is not a universal outcome. Instead, it depends on baseline diet, dietary exposure length, and sex.

## 1. Introduction

There is a worldwide epidemic of obesity and metabolic dysfunction [1]. The problem is worsening over time, and many studies reported increased obesity incidence after the COVID-19 pandemic [2]. While the new risk factors add to the growing problem, the main driver of the obesity pandemic is the worsening quality of our food, leading to increased caloric intake. This is mainly accounted for by an increase in sugar and fat consumption [3].While fat intake is a well-recognized risk factor, numerous studies have documented the detrimental health effects of sugar intake [4,5,6]. Conversely, reducing sugar consumption leads to improved metabolic health [7,8,9,10]. The majority of harmful effects of sugar have been attributed to its fructose component [11]. However, the detrimental metabolic outcomes of fructose intake are mainly observed in the context of co-ingestion with a high-fat diet (HFD). This brings into question whether fructose itself is sufficient to induce metabolic dysfunction or whether its effects are only observed in the setting of hypercaloric diets. This enigma is further exacerbated by the fact that fructose intake from fruits and vegetables appears to be protective from the development of metabolic dysfunction [12]. Thus, the question that remains to be answered is how the same molecule of fructose is associated with detrimental effects in one setting and beneficial effects in another.

One potential explanation acknowledges that fructose intake from solid food may be beneficial, while its consumption from sweetened drinks might be detrimental [13]. Indeed, increasing sugar content in a solid diet from 5 to 30% of calories had no effect on weight gain in mice [14]. When fructose is naturally found in solid foods, it comes together with fiber, vitamins, and antioxidants, which could lead to potential benefits by encouraging a feeling of fullness [15,16], slowing down its absorption to enhance protective intestinal fructose metabolism [17,18,19], and neutralizing its metabolic sequela by scavenging oxidative stress [20]. Conversely, fructose from sugar-sweetened beverages (SSBs) is rapidly absorbed and lacks these protective elements, potentially causing excessive hepatic fructose metabolism to promote the development of hepatic steatosis and insulin resistance [6,21,22,23]. While highly intuitive, this explanation fails to account for human studies showing that even liquid fructose intake leads to weight gain in some studies [24,25], while in others, it does not support weight gain [26,27]. More intriguingly, some studies documented that fructose intake worsens glucose homeostasis [28], while others do not substantiate this claim [29,30] or even indicate that fructose consumption can improve serum glucose and insulin response [31]. These contradictory findings have led to controversy in the field of fructose metabolism and undermine the importance of dietary guidelines recommending a reduction in sugar intake [32,33]. Our personal experience with this question came to the forefront when our chow-fed mice supplemented with 30% fructose solution failed to gain weight when our lab moved to Lexington, Kentucky, even though the same strain of mice in Boston, Massachusetts, consistently gained weight when provided fructose-sweetened water. These inconsistencies prompted the current study, which aims to explain how the background diets alter the metabolic consequences of fructose intake.

In this study, we set out to investigate how the intake of 30% fructose-sweetened water affects the development of obesity and metabolic complications in male and female mice fed different normal chow diets for different lengths of time.

## 2. Materials and Methods

### 2.1. Animals and Diets

In accordance with NIH guidelines, all animal protocols were approved by the IACUC of the University of Kentucky (2019-3289, 10 October 2019) and Joslin Diabetes Center, where different experiments were performed. At Joslin Diabetes Center, mice were housed in cages with corncob bedding at 20–22 °C on a 12 h light/dark cycle with ad libitum access to food and water. At the University of Kentucky, mice were housed in individually ventilated cages with woodchip bedding at 20–22 °C on a 14 h light/10 h dark cycle with ad libitum access to food and water. At both institutions, C57BL/6J male mice, 6–8 weeks of age, were obtained from Jackson Laboratory and assigned to different diets. These diets included the Boston chow diet (Lab Diet, #5020), which contains 3.46 kcal/g metabolizable energy and consists of 23% protein, 22% fat, and 55% carbohydrates, and the Lexington chow diet (Teklad, #2018), which includes 3.1 kcal/g metabolizable energy and consists of 24% protein, 18% fat, and 58% carbohydrates. The mice were given a 30% fructose solution in water or regular water and remained on the respective diets for ten weeks. Additionally, at the University of Kentucky, we tested the effects of 30% fructose intake on a semi-purified low-fat diet (LFD) for 10 weeks (Research Diets, #D12450K), which contains 3.85 kcal/g metabolizable energy, consisting of 20% protein, 10% fat, and 70% carbohydrates, followed by a crossover to the BCD for an additional 10 weeks. The animals were weighed, and their food and water intake were recorded once per week. GTT and ITT were performed after 8 or 18 weeks on the diets. Mice were sacrificed between 8:00 and 11:00 am. The mice were injected with saline or 1U of insulin (Humulin R-Lilly #HI-213) via inferior vena cava 10 min before tissue collection.

### 2.2. Glucose and Insulin Tolerance Tests

The glucose tolerance test (GTT) and insulin tolerance test (ITT) were used to assess in vivo glucose metabolism and insulin sensitivity. For the GTT, mice underwent overnight fasting for 12–16 h. Following fasting, glucose is administered intraperitoneally at a dose of 2 g/kg body weight. Blood glucose levels are measured at baseline (0 min) and at 15, 30, 60, and 120 min post-injection using a glucose meter (Infinity, US Diagnostics, Chicago, IL, USA), creating a time-course profile of glucose clearance. The area under the curve (AUC) has been calculated and is an index of glucose intolerance [34]. For ITT, insulin was administered to non-fasted mice intraperitoneally at a dose of 1 U/kg body weight. Blood glucose levels were measured at indicated times. The area above the cure (AAC) has been calculated and used as an index of insulin resistance. The AAC is not used in clinical practice to assess insulin sensitivity, but it is used in rodent research similar to AUC for OGTT [35].

### 2.3. Liver Histology and Hematoxylin and Eosin Staining (H and E)

Histology sections were prepared from formalin-fixed paraffin-embedded liver sections. H and E staining was performed by the University of Kentucky Pathology Research Core.

### 2.4. Oil Red O Staining (ORO)

Oil-Red O staining was performed to visualize hepatic neutral lipids. Frozen liver sections were air-dried for 30 min at room temperature and then fixed in 10% formalin for 15 min. After rinsing with PBS for 5 min, the slides were treated with 60% isopropyl alcohol for 5 min. Freshly prepared and filtered Oil-Red O stain was applied for 10 min. Slides were rinsed with 60% isopropyl alcohol for 2 min, followed by three washes in distilled water. A coverslip was mounted using warmed glycerol gelatin.

### 2.5. Measuring Quant Hepatic Triglycerides

Approximately 50 mg of liver tissue was weighed. Samples were kept frozen on dry ice to prevent degradation. Each tissue sample was placed in a 2 mL tube with a steel bead and 300 µL of chloroform–methanol (2:1, *v*/*v*) and homogenized with a tissue lyser at 1/25 oscillations for 5 min. After adding 700 µL of chloroform (final volume 1 mL), the samples were homogenized again and then centrifuged at 15 K for 15 min at 4 °C. A 10 µL aliquot of the supernatant was transferred to glass culture tubes (12 × 75 mm) and evaporated in a fume hood at room temperature for 1–4 h. Next, 200 µL of triglyceride reagent (Pointe Scientific, Canton, MI, USA, #T7535-500) was added to each tube and briefly vortexed. Next, 200 µL of reagent was added to the wells for the standard curve, which was prepared using standards (0, 1, 2, 4, 6, 8, and 10 µL of TG solution at 200 mg/dL). All samples were incubated for 5 min at 37 °C. Finally, 190 µL from each sample was transferred to a 96-well plate, and absorbance was read at 500 nm using a plate reader.

### 2.6. qPCR and mRNA Quantification

mRNA was extracted by homogenizing liver tissue in TRIzol (Invitrogen, Waltham, MA, USA, #15596018), treating it with chloroform, and precipitating in 70% ethanol. mRNA was purified using RNeasy Mini Kit columns (Qiagen, Hilden, Germany, #74106). cDNA was made using a High-Capacity cDNA Reverse Transcription Kit (Applied Biosystems, Waltham, MA, USA, #4368813). qPCR was performed utilizing C1000 Thermal Cycler (BioRad, Hercules, CA, USA, #CFX384) and QuantStudio^™^ 7 Flex Real-Time PCR System (TermoFisher Scientific, Waltham, MA, USA, #4485701). Primers used for qPCR are listed in Appendix A.

### 2.7. Protein Extraction and Immunoblot

Tissues were homogenized in RIPA buffer (EMD Millipore) with protease and phosphatase inhibitor cocktail (Bimake.com, Housto TX, USA, #B14002, #B15002). Proteins were separated using SDS-PAGE and transferred to the nitrocellulose membrane (BioRad, #1620112). Immunoblotting was achieved using the indicated antibodies listed in Appendix A. Images were captured by the ChemiDoc MP imaging system (BioRad, #12003154) and iBright imaging system (Thermo Fisher, #CL1000). Quantification of immunoblots was performed using ImageJ Version 1.54.

### 2.8. Plasma Insulin Levels and HOMA-IR Quantification

Following the manufacturer’s protocol, plasma insulin levels were measured using an ultra-sensitive mouse insulin ELISA kit (Crystal Chem, Elk Grove Village, IL, USA, #90080). Insulin resistance was assessed by calculating the Homeostasis Model Assessment of Insulin Resistance (HOMA-IR) index, using the following formula: HOMA-IR = (Fasting Insulin [μU/mL] × Fasting Glucose [mg/dL])/405 × 10. Specifically, the fasting glucose concentration was multiplied by the fasting insulin concentration, and the result was divided by 405, with the final value further multiplied by 10 to yield the HOMA-IR index.

### 2.9. Statistical Analyses

All statistical analyses and graph generation were conducted using GraphPad Prism version 10.3.0. Data are presented as mean ± SEM unless otherwise specified in the figure legends. The statistical tests for each data panel are detailed in the corresponding figure legends. Pairwise comparisons were conducted using Student’s *t*-test, and analyses involving multiple groups were performed using ANOVA followed by Tukey’s post hoc test. *p*-values < 0.05 were considered statistically significant.

## 3. Results

### 3.1. The Baseline Diet Modifies the Impact of Fructose on Weight Gain and Insulin Resistance

To determine the impact of high fructose intake on different baseline diets, we placed C57BL6/J male mice on two different chow diets. The 9F 5020 chow diet (Boston chow diet) contains 3.46% calories per gram and consists of 23% protein, 22% fat, and 55% carbs. The T 2018 chow diet (Lexington chow diet) contains 3.1% calories per gram and consists of 24% of protein, 18% of fat, and 58% of carbs (Appendix A). The mice were given a 30% fructose solution in water or regular water and remained on the respective diets for ten weeks (Appendix A). Mice on the Boston chow diet supplemented with fructose (BCD + F) gained more weight (36.5 ± 0.6 g) than mice on the Boston chow diet drinking regular water (BCD + W) (28.8 ± 1.4 g) (Figure 1A). Interestingly, mice on the Lexington chow diet supplemented with fructose (LXD + F) did not gain additional weight (27.9 ± 1.0 g) compared to the mice on the Lexington chow diet drinking regular water (LXD + W) (27.5 ± 0.9 g) (Figure 1B). In agreement with the weight gain, the BCD + F group consumed more calories (19.0 ± 1.90 kcal) than the mice on regular water (10.60 ± 0.74 kcal) (Figure 1C). However, the LXD + F group also consumed more calories (12.99 ± 0.30 kcal) than their water-supplemented counterparts (10.11 ± 0.13 kcal) (Figure 1D). Examining the sources of these calories, we found that mice given fructose water drank significantly more than those provided regular water (6.74 ± 0.50 mL vs. 3.80 ± 0.22 mL) (Appendix A). This increase in fructose intake did not affect solid food consumption in the BCD + F group compared to the BCD + W group (3.34 ± 0.41 g vs. 3.04 ± 0.21 g) (Appendix A). When the source of calories was further subdivided into protein, fat and carbohydrate, mice in the BCD + F group consumed more calories from carbohydrates, while the caloric intake from protein and fat was unchanged (Appendix A). On the other hand, the mice on LXD also drank more fructose than regular water (4.84 ± 0.13 mL vs. 3.32 ± 0.14 mL) (Appendix A), but, interestingly, solid food intake was significantly reduced in the LXD + F-fed mice compared to the LXD+W-fed mice (2.50 ± 0.05 g vs. 3.30 ± 0.04 g) (Appendix A). Correspondingly, mice in the LDX+F group consumed more calories from carbohydrates, but they showed a decrease in calories from protein and fat. A decrease in solid food intake in the LXD-, but not in the BCD-fed mice may account for the observation that these mice did not gain weight in response to fructose.

Next, we assessed glucose handling in these mice. Fasting insulin levels were not significantly altered in the BCD + F group (2.40 ± 0.21 ng/mL) compared to the BCD + W group (2.32 ± 0.40 ng/mL) (Figure 1E). Fasting blood glucose tended to be elevated in the BCD + F group (150.7 ± 9.6 ng/mL) compared to the BCD + W-fed mice (135.1 ± 9.9 ng/mL) without reaching statistical significance (Figure 1F). The Homeostatic Model Assessment for Insulin Resistance (HOMA-IR) was not altered in the BCD + F-fed mice compared to the BCD + W-fed mice (8.90 ± 0.84 ng/mL vs. 8.08 ± 1.72 ng/mL) (Figure 1G). On the other hand, the mice on the LXD + F diet (0.9 ± 0.2 ng/mL) had significantly lower serum insulin levels than those in the LXD + W group (1.8 ± 0.3 ng/mL) (Figure 1H). They also had reduced fasting blood glucose levels (118 ± 5.4 mg/dL vs. 132.3 ± 3.4 mg/dL) (Figure 1I), and the LXD + F-fed group had lower HOMA-IR values (2.5 ± 0.5 ng/mL) compared to the LXD + W group (5.8 ± 1.0 ng/mL) (Figure 1J).

In line with these findings, the glucose tolerance test (GTT) area under the curve (AUC) indicated no significant differences between the BCD + F- and the BCD + W-fed mice (23,655 ± 1133 mg/dL vs. 25,134 ± 2171 mg/dL) (Figure 1K,L). However, a trend for impaired insulin tolerance tests (ITT) (*p* = 0.09) was observed in the area above the curve (AAC) for the BCD + F-fed group compared to the BCD + W-fed control (2348 ± 586.5 mg/dL vs. 3534 ± 300.0 mg/dL) (Figure 1M,N). Moreover, the blood glucose level at 120 min of ITT was significantly elevated in BCD + F-fed mice compared to BCD + W-fed mice (Figure 1M). On the other hand, on LXD, the AUC for the GTT showed no significant differences between the LXD + F and LXD + W groups (33,003 ± 968.5 ng/mL vs. 31,288 ± 1054 ng/mL) (Figure 1O,P). Similarly, the AAC from the ITT did not demonstrate significant alterations between the LXD + F and LXD + W groups (4766 ± 317.0 ng/mL vs. 4187 ± 466.6 ng/mL), although blood glucose was better in the LXD + F group at 15 min compared to the LXD + W group (Figure 1Q,R).

To evaluate the hepatic insulin signaling, we injected 1 unit/mouse of insulin or saline into the inferior vena cava 10 min before sacrifice. The BCD + F group showed significantly decreased Akt phosphorylation compared to the BCD + W control group (Figure 1S,T). In contrast, the insulin-stimulated Akt phosphorylation in the LXD + F group was not impaired compared to the LXD + W control group (Figure 1S,T). Furthermore, all mice on LXD had higher Akt phosphorylation than the mice on BCD. ERK phosphorylation increased across all mice treated with insulin, irrespective of the specific dietary composition. ERK phosphorylation was significantly lower in all mice fed with the LXD compared to the BCD (Figure 1S,T), in agreement with the insulin levels. In summary, fructose supplementation increased weight gain, impaired insulin tolerance at 120 min, and reduced hepatic insulin signaling on the BCD, while fructose intake on LXD did not induce weight gain and even improved some measures of insulin resistance.

### 3.2. Fructose Supplementation on BCD but Not on LXD Supports Hepatic Steatosis

To explain the differences in insulin resistance, we next investigated liver physiology. The liver weight of mice on the BCD supplemented with fructose was significantly higher compared to mice on regular water (1.60 ± 0.06 vs. 1.30 ± 0.07) (Figure 2A). Conversely, the liver weight of mice on the LXD + F did not differ from mice on the LXD + W (1.20 ± 0.10 vs. 1.20 ± 0.05) (Figure 2B). In agreement with the liver weight, BCD + F-fed mice developed hepatic steatosis compared to BCD + W-fed mice, as assessed by Oil Red O (Figure 2C) and Hematoxylin and Eosin (H and E) staining (Appendix A). Similarly, liver triglyceride (TG) accumulation was higher in the BCD + F group compared to the BCD + W group (Figure 2D). In contrast, fructose supplementation in the LXD-fed mice did not affect steatosis or TG accumulation compared to the LXD + W-fed mice (Figure 2C,E).

Next, we investigated whether the difference in steatosis could be explained by the differences in hepatic fructose metabolism. The expression of Khk-c, the isoform of ketohexokinase that is primarily involved in the first step of fructolysis, was unchanged in BCD + F-fed mice compared to BCD + W-fed mice. Similarly, fructose supplementation in the BCD group did not alter the expression of Khk-a, the alternative isoform of ketohexokinase that plays a role in regulating fructose metabolism in different tissues (Figure 2F). However, there was a small but significant increase in Khk-c expression in the LXD + F-fed mice compared to the LXD + W-fed mice. Furthermore, there was a trend towards an increase in Khk-a expression in the LXD + F group, as determined by quantitative PCR (qPCR) (Figure 2G). We previously showed that fructose intake can alter protein levels via post-translational modifications, resulting in a poor correlation between protein and mRNA expression [36,37]. Indeed, fructose supplementation in the BCD group resulted in elevated KHK-C protein levels compared to the BCD + W group (Figure 2H, Appendix A). Similarly, fructose supplementation in the LXD group led to an increase in KHK-C protein levels compared to the LXD + W group. Moreover, there was a significantly greater fold induction of KHK-C with fructose on BCD than on LXD when normalized to baseline expression in the water group for each diet (Appendix A). KHK-C protein in the liver is about 300 times more abundant than KHK-A [38,39]. Accordingly, we only detected a faint KHK-A band, which was not affected by fructose supplementation of either BCD or LXD (Figure 2H, Appendix A). In summary, fructose intake in BCD more strongly induced KHK-C protein than in LXD, which was accompanied by greater steatosis in the BCD + F- compared to the LXD + F-fed mice.

Fructose consumption strongly stimulates hepatic de novo lipogenesis (DNL) [40]. In the BCD-fed mice, fructose supplementation led to increased hepatic mRNA levels of enzymes involved in fatty acid synthesis, including ATP citrate lyase (Acly) and acetyl-CoA carboxylase 1 (Acc1), compared to BCD + W-fed mice (Figure 2I). Additionally, there was a trend toward increased fatty acid synthase (Fasn) expression, while the expression of stearoyl-CoA desaturase 1 (Scd1) remained unchanged. On the other hand, in the LXD-fed mice, fructose supplementation elevated hepatic mRNA levels of Acly, Acc1, and Scd1 and tended to increase Fasn levels compared to LXD + W-fed mice (Figure 2J). The protein levels of ACLY, ACC1, FASN, and SCD1 were significantly increased in the BCD + F group compared to the BCD + W group (Figure 2K, Appendix A). Similarly, the protein levels of ACLY, ACC1, FASN, and SCD1 were significantly elevated in the LXD + F-fed mice compared to the LXD + W-fed mice. Consistent with the greater fat content of BCD than LXD, baseline protein levels of DNL enzymes were lower in the BCD compared to the LXD. This resulted in greater fold induction of the DNL proteins ACLY, ACC1, FASN, and SCD1 with fructose supplementation of the BCD than on the LXD (Appendix A), in agreement with greater KHK-C induction in this group.

Fructose intake can also reduce fatty acid β-oxidation (FAO). Fructose supplementation did not alter hepatic mRNA expression levels of key enzymes involved in FAO, including carnitine palmitoyl transferase 1A (Cpt1a), acyl-CoA oxidase 1 (Acox1), and acyl-CoA thioesterase 1 (Acot1) on either BCD (Appendix A) or LXD (Appendix A). However, the protein level of CPT1A, the rate-limiting enzyme of FAO, was significantly reduced in the BCD + F-fed mice compared to the BCD + W-fed mice (Figure 2L, Appendix A), whereas the protein levels of Acox1 and Acot1 remained unchanged. Likewise, in the LXD group, fructose supplementation led to a significant reduction in the protein level of CPT1A, with no significant changes observed in the levels of Acox1 and Acot1 when compared to LXD + W-fed mice (Figure 2L, Appendix A). Additionally, the levels of other enzymes involved in fatty acid β-oxidation, including carnitine acylcarnitine translocase (CACT), organic cation/carnitine transporter 2 (OCTN2), carnitine palmitoyl transferase II (CPT2), acyl-CoA dehydrogenase long chain (ACADL), acyl-CoA dehydrogenase very long chain (ACADVL), and hydroxy acyl-CoA dehydrogenase trifunctional multienzyme complex subunit alpha (HADHA) were minimally affected with fructose supplementation in either BCD or LXD groups (Appendix A). In summary, fructose supplementation supports the development of hepatic steatosis only in mice on the BCD but not in mice on the LXD. On a molecular level, fructose showed a stronger fold induction in the protein levels of KHK-C and the enzymes mediating the DNL pathway on the BCD compared to the LXD.

### 3.3. Switching from a Low-Fat Diet to a Boston Chow Diet Restores Weight Gain and Insulin Resistance

To test if the outcomes of fructose supplementation are solely dictated by the underlying baseline diet, we tested the effects of fructose intake on a semi-purified low-fat diet (LFD) for 10 weeks, followed by a crossover to the BCD for an additional 10 weeks (Appendix A). The LFD (D12492) provides 3.85 calories per gram, consisting of 20% protein, 10% fat, and 70% carbohydrates (Appendix A). Mice on the low-fat diet supplemented with fructose (LFD + F) exhibited a non-significant trend towards lower weight gain (30.0 ± 0.6 g) compared to those on the low-fat diet consuming regular water (LFD + W) (31.8 ± 1.1 g) (Figure 3A). Interestingly, upon switching to the BCD, fructose supplementation resulted in a significant increase in weight gain (43.1 ± 1.0 g) compared to the BCD + W-fed mice (37.5 ± 2.0 g) (Figure 3B). Total caloric intake in the LFD + F group was not significantly changed (11.80 ± 0.20 kcal) than in the mice on regular water (11.70 ± 0.22 kcal) (Figure 3C); however, mice fed the BCD + F diet had a significant increase in caloric intake (14.01 ± 0.44 kcal) compared to the BCD + W group (12.20 ± 0.13 kcal) (Figure 3D). Moreover, mice given fructose on LFD drank significantly more than those provided regular water (4.34 ± 0.20 mL vs. 2.93 ± 0.05 mL) (Appendix A). In contrast, solid food consumption was significantly reduced in the LFD + F group compared to the LFD+W group (1.82 ± 0.07 g vs. 3.03 ± 0.05 g) (Appendix A). The mice on BCD also drank more fructose than regular water (4.30 ± 0.11 mL vs. 3.30 ± 0.07 mL) (Appendix A). Similar to the LFD groups, solid food intake was significantly decreased in the BCD + F-fed mice compared to the BCD + W-fed mice (2.60 ± 0.04 g vs. 3.52 ± 0.03 g) (Appendix A). However, the decrease in solid food intake was greater on LFD (41%) than on BCD (28%). This indicates that fructose-supplemented mice experienced a relative increase in solid food intake when switched from LFD to BCD, which in part explains higher weight gain.

Fasting insulin level was markedly elevated in the LFD + F group compared to the LFD+W group (0.52 ± 0.10 ng/mL vs. 0.22 ± 0.04 ng/mL) (Figure 3E). Moreover, the LFD + F group showed an increase in fasting blood glucose levels compared to the LFD+W-fed mice (164.1 ± 6.94 mg/dL vs. 140.3 ± 6.90 mg/dL) (Figure 3F), and HOMA-IR revealed a trend towards increased insulin resistance (2.13 ± 0.60 ng/mL vs. 0.80 ± 0.15 ng/mL) (Figure 3G). Similarly, serum insulin levels in the BCD + F group (1.80 ± 0.23 ng/mL) were significantly higher than those in the BCD + W group (0.90 ± 0.35 ng/mL) (Figure 3H). However, the BCD + F diet did not significantly alter fasting blood glucose levels (161.0 ± 9.44 mg/dL vs. 151.9 ± 7.30 mg/dL) (Figure 3I), and HOMA-IR values remained unchanged between the BCD + F-fed mice (7.01 ± 1.20 ng/mL) and the BCD + W-fed mice (4.11 ± 1.40 ng/mL) (Figure 3J).

The IPGTT revealed significantly higher AUC in the LFD + F group compared to the LFD+W group (34,359 ± 1268 ng/mL vs. 27,788 ± 1064 mg/dL) (Figure 3K,L), and the glucose levels during GTT were significantly higher in the LFD + F group at 15, 30, and 60 min compared to the LFD+W group (Figure 3K). Conversely, the ITT demonstrated no significant difference in the AAC between the LFD + F and LFD + W groups (3084 ± 370 ng/mL vs. 2253 ± 312 mg/dL) (Figure 3M,N). On the other hand, the GTT AUC indicated a trend towards impaired glucose tolerance in BCD + F-fed mice compared to the BCD + W-fed mice (51,875 ± 3072 ng/mL vs. 42,713 ± 3289 mg/dL; *p* = 0.06) (Figure 3O,P), and the glucose levels were significantly higher in the BCD + F group at 60 and 120 min compared to the BCD + W group (Figure 3O). The overall ITT AAC showed no significant difference (5567 ± 443 mg/dL vs. 6573 ± 430 mg/dL), but glucose levels were higher in the BCD + F-fed mice compared to the BCD + W-fed controls at 60 and 90 min (Figure 3Q,R).

The assessment of hepatic insulin signaling revealed that fructose supplementation in the LFD group significantly reduced Akt phosphorylation compared to the LFD + W control group (Figure 3S,T). Similarly, fructose supplementation of the BCD-fed mice resulted in a significant reduction in the insulin-stimulated Akt phosphorylation compared to the BCD + W-fed control. Overall, Akt phosphorylation was higher in all LFD-fed mice compared to those on BCD. Furthermore, ERK phosphorylation increased in all mice treated with insulin, regardless of the dietary composition (Figure 3S,T). In summary, fructose supplementation led to increased weight gain when mice were transitioned from the LFD to the BCD. Interestingly, in both the LFD and BCD groups, fructose consumption impaired some aspects of glucose tolerance and insulin resistance compared to their respective controls. Moreover, in LFD-fed mice, adverse changes in glycemic status with fructose ingestion were observed despite no significant increase in body weight.

### 3.4. Fructose Enhances Hepatic Steatosis More Strongly in Mice on BCD than on LFD

Liver weight was minimally increased in mice consuming an LFD supplemented with fructose compared to those provided with regular water (1.40 ± 0.04 g vs. 1.09 ± 0.09 g) (Figure 4A). However, mice on the BCD supplemented with fructose demonstrated a greater increase in liver weight relative to those on regular water (2.01 ± 0.20 g vs. 1.60 ± 0.10 g) (Figure 4B). Despite a slight increase in liver weight, the LFD + F group did not exhibit an increase in hepatic steatosis compared to the LFD + W group, as evidenced by the ORO (Figure 4C) and H and E staining (Appendix A). Consistent with the higher liver weight, the BCD + F group developed hepatic steatosis compared to the BCD + W group (Figure 4C; Appendix A). In agreement with steatosis, hepatic TG accumulation was not affected in the LFD + F group compared to the LFD + W control (Figure 4D). In contrast, liver TG levels were elevated in the BCD + F group relative to the BCD + W control (Figure 4E).

Next, we assessed the effects of fructose on KHK levels in mice on these diets. The expression of Khk-c and Khk-a remained unchanged in the LFD + F-fed mice relative to the LFD + W-fed controls. (Figure 4F). In contrast, Khk-c and Khk-a expression was upregulated in the BCD group following fructose supplementation compared to the BCD + W group (Figure 4G). At the protein level, fructose supplementation in the LFD group resulted in the elevated protein levels of KHK-C compared to the LFD + W group (Figure 4H; Appendix A). Similarly, KHK-C protein levels were significantly elevated in the BCD + F-fed mice relative to the BCD + W-fed counterparts. Calculating fold change over its baseline, fructose resulted in greater induction of KHK-C on BCD compared to LFD (Appendix A). Conversely, fructose supplementation did not induce a significant change in the KHK-A protein levels in either the LFD or BCD group (Figure 4H; Appendix A).

In the LFD-fed mice, fructose supplementation did not elicit significant changes in the hepatic mRNA levels of key enzymes involved in fatty acid synthesis, including Acly, Acc1, Fasn, and Scd1, when compared to the LFD + W-fed controls (Figure 4I). In contrast, fructose supplementation in the BCD-fed mice led to a significant upregulation of hepatic mRNA levels of Acly, Acc1, Fasn, and Scd1 relative to the BCD + W-fed mice (Figure 4J). At the protein level, the LFD + F group exhibited increased levels of ACLY, ACC1, and FASN compared to the LFD + W group (Figure 4K; Appendix A). Similarly, in the BCD group, fructose supplementation resulted in a significant elevation of protein levels for ACLY, ACC1, FASN, and SCD1 compared to the BCD + W-fed control (Figure 4K; Appendix A). Moreover, the BCD had lower baseline levels of DNL enzymes; hence, fructose supplementation resulted in a greater fold increase in DNL enzymes in mice on BCD than on LFD (Appendix A). This is consistent with the greater induction of KHK-C protein in the mice on the BCD than on the LFD.

In both LFD- and BCD-fed mice, fructose supplementation did not markedly affect the hepatic mRNA levels of enzymes involved in fatty acid beta-oxidation, including Cpt1a, Acox1, and Acot1, when compared to regular water controls (Appendix A). At the protein level, CPT1A was not affected in the LFD + F-fed mice compared to their LFD + W-fed counterparts (Figure 4L; Appendix A). Conversely, CPT1A was significantly decreased with fructose supplementation in the BCD group compared to the BCD + W group. In contrast, ACOX1 and ACOT1 protein levels remained unchanged with fructose supplementation on either diet (Figure 4L; Appendix A). In summary, fructose supplementation on BCD compared to the LFD enabled greater induction in KHK-C and DNL enzymes while lowering the CPT1A protein levels, which may explain the development of hepatic steatosis in these mice.

### 3.5. Fructose Supplementation Results in Weight Gain and Insulin Resistance in Male but Not Female Mice

In the subsequent experiments, we compared the effects of fructose supplementation in male and female mice. Like the previous experiment, the mice were on LFD for ten weeks and then crossed over to BCD for another ten weeks (Appendix A). As previously shown, male mice gained weight with fructose supplementation when crossed to BCD. In contrast, female mice did not gain weight with fructose supplementation on LFD (19.7 ± 0.6 g vs. 20.9 ± 0.6 g), and similarly, their weight gain remained unchanged when switched to BCD compared to female mice receiving regular water (23.6 ± 0.8 g vs. 25.5 ± 1.4 g) (Figure 5A). In agreement with body weight, the mass of perigonadal (PG) adipose tissue increased in male mice supplemented with fructose, while it was unchanged in female mice (Appendix A). As expected, both male (4.30 ± 0.11 mL vs. 3.30 ± 0.07 mL) and female (3.44 ± 0.16 mL vs. 2.92 ± 0.02 mL) mice drank more fructose than regular water (Figure 5B). This was associated with a corresponding reduction in solid food intake in both male (2.60 ± 0.04 g vs. 3.52 ± 0.03 g) and female mice (2.18 ± 0.11 g vs. 4.30 ± 0.10 g) after 20 weeks on the diets (Figure 5C). However, female mice exhibited a greater reduction in solid food intake with fructose than males (50% vs. 28%). While the caloric intake from fat and protein was lower in both male and female mice supplemented with fructose, total carbohydrate intake was higher in males given fructose, but it was actually unchanged in female mice supplemented with fructose (Appendix A). Consequently, the total caloric intake significantly increased in male mice consuming fructose (14.01 ± 0.44 kcal vs. 12.20 ± 0.13 kcal), while it decreased in female mice given fructose (11.40 ± 0.30 kcal vs. 14.71 ± 0.40 kcal) relative to the regular water group (Figure 5D).

Unlike males, fasting insulin levels were not increased with fructose feeding in female mice (0.13 ± 0.03 ng/mL vs. 0.11 ± 0.02 ng/mL), and insulin levels were much lower in females compared to male mice (Figure 5E). Blood glucose levels (Figure 5F) and HOMA-IR (Figure 5G) were not significantly affected in male or female mice exposed to fructose for 20 weeks. However, female mice had significantly lower blood glucose and HOMA-IR values than males. Similarly, female mice supplemented with fructose showed no significant change in glucose tolerance AUC (29,449 ± 1917 ng/mL vs. 32,858 ± 3285 mg/dL) compared to those given regular water (Figure 5H,I), even though males showed a tendency to have impaired glucose tolerance with fructose intake. Female mice also demonstrated no significant difference in ITT AAC between the fructose and regular water groups (8106 ± 258 ng/mL vs. 6472 ± 1001 mg/dL), even though blood glucose values tended to be lower at 30, 60, and 90 min in fructose-fed female mice (Figure 5J,K). On the other hand, glucose values at 60 and 90 min were actually higher in male mice exposed to fructose.

When assessing the hepatic insulin signaling, the insulin-stimulated AKT phosphorylation remained unaltered in female mice supplemented with fructose compared to the regular water group (Figure 5L,M), despite a significant reduction in male mice. However, like the males, female mice given fructose exhibited a slight decrease in ERK phosphorylation compared to their water controls. In summary, fructose-fed female mice displayed lower weight gain and were protected from glucose intolerance and insulin resistance compared to male mice.

### 3.6. Fructose Supplementation in Female Mice More Strongly Upregulates KHK-C and DNL Enzymes than in Males

Similar to males, female mice supplemented with fructose showed a trend toward an increase in liver weight compared to those receiving regular water (1.40 ± 0.09 g vs. 1.20 ± 0.08 g) (Figure 6A). When corrected for body weight, fructose-fed female mice demonstrated a significant increase in the liver-to-body weight ratio (5.50 ± 0.20 g vs. 4.53 ± 0.20 g) compared to female mice on regular water (Figure 6B). Consistent with liver weight, fructose supplementation in male mice led to the development of hepatic steatosis. Similarly, female mice given fructose also developed hepatic steatosis, as indicated by ORO staining (Figure 6C) and H and E staining (Appendix A). Liver TGs were significantly increased in female mice given fructose compared to water controls (47.21 ± 2.20 g vs. 19.09 ± 2.30 g), similar to an increase observed in fructose-fed male mice (Figure 6D). On the other hand, the hepatic cholesterol levels were profoundly increased in female mice given fructose compared to water controls (3.70 ± 0.20 g vs. 2.70 ± 0.11 g), while fructose-fed male mice only showed a slight and non-significant trend in hepatic cholesterol (2.80 ± 0.22 g vs. 2.31 ± 0.11 g) (Figure 6E). In summary, female mice supplemented with fructose develop a much larger increase in liver-to-body weight ratio, hepatic steatosis, and hepatic cholesterol levels than males.

Similar to that in males, the KHK-C protein levels increased in female mice supplemented with fructose compared to their counterparts in regular water (Figure 6F,H; Appendix A). When corrected to their respective baseline levels in water groups, the fold change increase in KHK-C with fructose intake was stronger in female than in male mice (Figure 6H). In contrast, the protein levels of KHK-A did not change in male or female mice supplemented with fructose (Figure 6F,H; Appendix A). Furthermore, the protein levels of ACLY, ACC1, FASN, and SCD1 increased with fructose supplementation in both male and female mice (Figure 6G). Again, fold change increase with fructose supplementation was stronger for ACLY and ACC1 in females compared to male mice (Figure 6G,H; Appendix A).

Similar to male mice, female mice supplemented with fructose exhibited a significant reduction in the CPT1A protein levels compared to those receiving regular water (Figure 6I,J). Conversely, Acox1 and Acot1 protein levels showed no significant alterations in either male or female mice following fructose supplementation (Figure 6I,J). In summary, fructose supplementation in both male and female mice resulted in a significant increase in KHK-C protein levels, along with elevated levels of ACLY, ACC1, FASN, and SCD1 proteins. The increase in KHK-C and DNL enzymes was stronger in female mice, in agreement with a higher liver-to-body weight ratio and liver steatosis.

## 4. Discussion

Our study demonstrated that the metabolic effects of fructose intake in mice vary significantly based on the external and internal variables that accompany its consumption. Indeed, male mice drinking 30% fructose solution on the Boston chow diet for ten weeks gain weight, develop hepatic steatosis, and have impaired hepatic insulin signaling. Remarkably, the mice consuming the same concentration of fructose on the Lexington chow diet do not gain additional weight, have normal lipid content in the liver, and remain insulin sensitive. To investigate whether altering the baseline diet is sufficient to drive these differences, we placed fructose-supplemented mice on a low-fat diet for ten weeks, then crossed them over to BCD for another ten weeks. The mice on LFD do not gain weight or develop hepatic steatosis when exposed to fructose, but they exhibit hepatic insulin resistance. Indeed, once switched to the BCD, the mice gain significantly more weight, exhibit greater lipid accumulation in the liver, and show a deterioration of the hepatic insulin signaling. The worsening of metabolic dysfunction is also influenced, in part, by the length of the diet exposure, as these metabolic parameters worsened between ten and twenty weeks on the diets. In addition, the response to fructose intake is driven by sex differences. While fructose-fed male mice develop metabolic dysfunction, female mice fed fructose for twenty weeks do not gain weight and remain insulin-sensitive despite accumulating a large amount of lipids in the liver. Together, these findings suggest that fructose-induced metabolic dysfunction is not a universal outcome but is highly dependent on the baseline diet, the length of the dietary exposure, and sex.

Our findings indicate that the baseline diet strongly modulates the metabolic response to fructose. In terms of weight gain, fructose supplementation resulted in significant weight gain when consumed on BCD. However, there was no weight gain with fructose intake on LXD, and mice tended to weigh less when given fructose on LFD. Moreover, the same mice that did not gain weight when fed fructose on LFD gained weight when the baseline diet was switched to BCD. The response to fructose supplementation could be, in part, explained by the carbohydrate load of the baseline diet. When mice consume fructose together with a carbohydrate-rich LFD, their solid food intake decreases the most. Solid food intake is also lower with fructose consumption on the LXD, which contains fewer carbohydrates than the LFD. However, the reduction in solid food intake is less pronounced on the LXD compared to the LFD. Interestingly, there was no reduction in solid food intake when fructose was consumed on the BCD, which is relatively high in fat. Therefore, the carbohydrate load in the solid diet is critical in determining how liquid fructose affects the intake of the solid diet [41]. Indeed, Solon-Biet et al. found that sweet preference for sugar water depends on the carbohydrate balance. Mice on a high-carbohydrate diet drank less sucrose, while those on a low-carbohydrate diet consumed more sucrose [42]. The studies of nutritional geometry also indicate that the protein content of the diet drives energy intake from a solid diet [43]. All three of our diets, BCD, LXD, and LFD, contain similar amounts of protein (23, 24, and 20% protein, respectively). Therefore, it is unlikely that the protein content affected the fructose-adjusted solid food intake. If anything, the mice prefer the average protein intake to be 23% [43]. Thus, it would be expected that they would eat more, not less, of LFD to keep the optimal protein intake when supplemented with fructose. In summary, drinking fructose on a low-fat, high-carbohydrate diet triggers a compensatory decrease in solid food intake and prevents the development of metabolic complications. This is akin to the protective effects of fructose when consumed from fruits and vegetables as a part of a well-balanced low-fat diet, while consumption of sugar-sweetened beverages on a standard American diet is harmful to metabolic health [44].

Fructose metabolism via KHK-C strongly induces lipogenesis [45,46]. Our results demonstrate that the baseline diet also influences the propensity of fructose to induce hepatic steatosis. Indeed, fructose induces hepatic triglyceride accumulation when consumed on a BCD but not on a LXD or LFD. These differences can be explained by the propensity of the baseline diet to suppress KHK-C and the endogenous DNL pathways. Indeed, diets high in fat lower the endogenous DNL [47]. The BCD has the highest amount of fat and, therefore, the lowest percentage of carbohydrates. Thus, the baseline levels of KHK-C and DNL proteins were the lowest on BCD, and they experienced the greatest fold-induction with fructose supplementation. On the other hand, the baseline protein levels of KHK-C and DNL enzymes were the highest on LFD since this diet has the lowest fat content. Upon the addition of fructose to LFD, there is only a marginal increase in KHK-C and DNL enzymes. Thus, fructose increases KHK-C and the endogenous DNL pathway in all diets, but the diets that have high fat content and, therefore, low baseline lipogenesis exhibit the greatest fold increase in lipogenesis with fructose supplementation. On the other hand, fructose supplementation also reduces the protein levels of CPT1A, the rate-limiting enzyme of FAO, in all three diets. Since the BCD contains the highest percent of calories from fat, it is understandable that fructose intake would have the largest effect on this diet. These findings support the well-established role of fructose in reducing FAO [48]. Another interesting aspect of our research is that it documents that fructose intake had a larger effect on the protein levels of DNL and FAO enzymes rather than on their mRNA expression. This observation is in line with our prior studies documenting that fructose via KHK-C affects the post-translational protein modifications and, thus, the protein stability [36,37]. Taken together, our results show that consumption of fructose along with a diet that contains the largest amount of fat supports the development of hepatic steatosis. Our study was not designed to determine the minimal amount of dietary fat required to induce fructose derangements, but this appears to be greater than twenty percent. Moreover, fructose induces KHK-C and hepatic steatosis in all conditions tested. However, steatosis does not always correlate with weight gain or hepatic insulin resistance. Therefore, targeting KHK-C for the management of metabolic dysfunction-associated steatotic liver disease (MASLD) is predicted to be beneficial for patients with or without obesity and in both sexes.

Sex is another biological variable that profoundly affects the propensity of fructose to induce metabolic complications. Male mice supplemented with fructose gain weight, develop impaired glucose tolerance, and develop hepatic steatosis. Female mice, on the other hand, do not gain additional weight and remain glucose-tolerant when exposed to fructose. This is in agreement with a generally lower risk of insulin resistance [49] and fatty liver disease [50,51] in women compared to men. Even though female mice do not gain additional weight when exposed to fructose, they have a larger induction in KHK-C and develop profound hepatic steatosis. This indicates that female mice are better able to compensate for the detrimental effects of fructose rather than that they are protected from developing fructose-induced complications. This is also true in human subjects, where females had a greater fructose-induced FGF21 response compared to males [52]. Indeed, the propensity of female mice to gain weight increases following ovariectomy [53] or housing in a thermoneutral environment [54]. Thus, it remains to be determined if fructose intake in female subjects could lead to full-blown metabolic complications when combined with other metabolic risk factors.

## 5. Limitations

The limitations of this study include utilizing pure fructose-sweetened drinks when the majority of SSBs are sweetened with sucrose or high-fructose corn syrup, which is a mixture of fructose and glucose. Indeed, Wali et al. reported that the worst metabolic outcomes occur when drinks contain a 50:50 fructose to glucose mixture [55,56]. Moreover, FGF21 is robustly increased with sugar intake, while exogenous FGF21 administration reduces sugar consumption [57,58]. We did not investigate FGF21 biology in our study. Lastly, the microbiome metabolizes the excess fructose to acetate, which circulates to the liver to support DNL [59]. The limitation of our study is that we did not investigate the microbiome effects on the propensity of fructose to induce metabolic dysfunction on different diets.

## 6. Conclusions

This study demonstrates that the metabolic effects of fructose are highly dependent on baseline diet, length of dietary exposure, and sex as a biological variable. Under all conditions tested, fructose intake increased KHK-C and activated the endogenous DNL pathway. However, weight gain and hepatic insulin resistance did not universally follow high fructose intake. These findings highlight the complexity of fructose metabolism on metabolic health, emphasizing the need to consider both the underlying diet and patient-specific risk factors.

Moreover, our study explains the conundrum of early studies on fructose metabolism reporting positive outcomes on glucose and insulin response [31]. As such, research in the 80s and 90s entertained the possibility that fructose may be used as an alternative sweetener for patients with diabetes [60]. Indeed, we observed the beneficial effects of fructose on insulin resistance when fructose is consumed on LXD. As the quality of our diet continued to deteriorate, the effects of fructose on metabolic outcomes have largely turned out to be undesirable. Indeed, we observe this when fructose is consumed on BCD. Thus, the propensity of fructose to induce metabolic dysfunction is largely dependent on the underlying baseline diet.

## 7. Future Perspectives and Directions

In future research, we plan to incorporate microbiota studies to investigate its possible role in mediating fructose-induced metabolic dysfunction. Additionally, changes in the microbiota could influence the hepatic ketohexokinase activity and de novo lipogenesis, the key pathways identified in this study.

## Figures and Tables

**Figure 1 nutrients-17-00124-f001:**
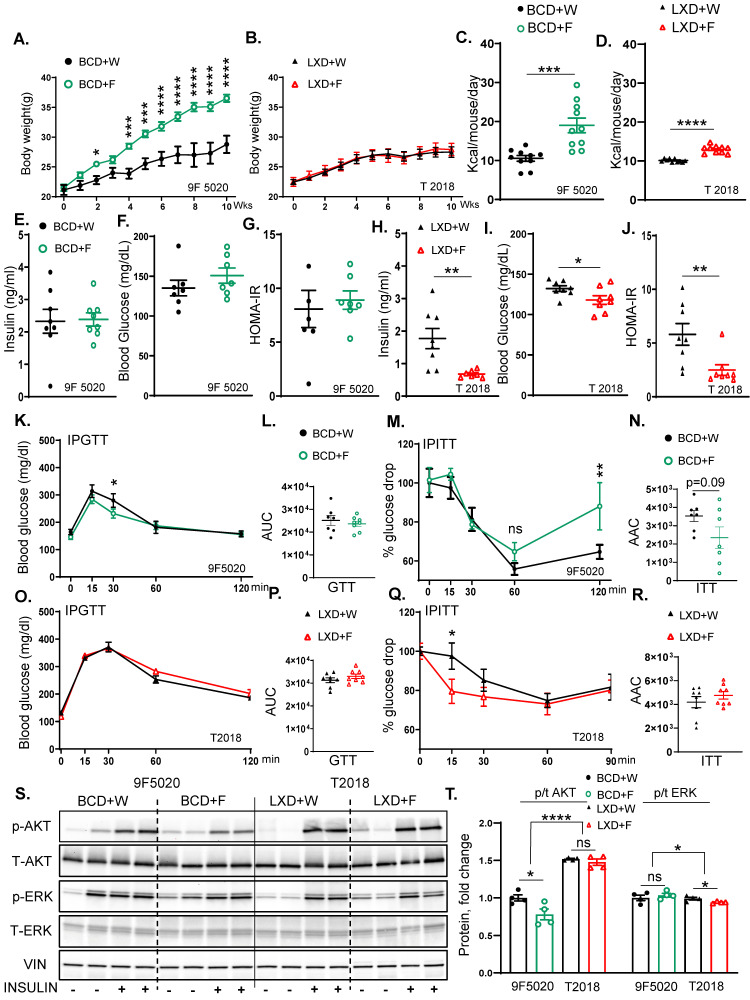
The baseline diet modifies the impact of fructose on weight gain and insulin resistance. (**A**) Weight gain of male mice on Boston chow diet (9F5020) supplemented with either regular water (BCD + W) or 30% fructose-sweetened water (BCD + F) for 10 weeks (*n* = 8 mice per group). (**B**) Weight gain of male mice on Lexington chow diet (T2018) supplemented with either regular water (LXD + W) or 30% fructose-sweetened water (LXD + F) for 10 weeks (*n* = 8 mice per group). (**C**) Average total caloric intake in mice on BCD (**D**) and LXD. (**E**) Fasting insulin levels, (**F**) blood glucose concentrations, and (**G**) Homeostatic Model Assessment for Insulin Resistance (HOMA-IR) for BCD-fed mice at 8 weeks on diet. (**H**) Fasting insulin levels, (**I**) blood glucose concentrations, and (**J**) HOMA-IR for LXD-fed mice at 8 weeks on the diet. (**K**) Glucose tolerance test (GTT) for BCD-fed mice performed after 8 weeks and (**L**) area under the curve (AUC) analysis for GTT results in BCD-fed mice. (**M**) Insulin tolerance test (ITT) in BCD-fed mice at 8 weeks, and (**N**) area above the curve (AAC) for ITT in BCD-fed mice. (**O**) GTT results for LXD-fed mice at 8 weeks and (**P**) AUC analysis for GTT. (**Q**) ITT in LXD-fed mice at 8 weeks on the diet and (**R**) AAC for ITT in LXD-fed mice. (**S**) Western blot analysis of phosphorylated and total AKT and ERK proteins in BCD and LXD-fed mice (*n* = 4 mice/group) and (**T**) Image J quantification of Western blot results for phosphorylated and total AKT and ERK proteins. Statistical comparisons were conducted using Student’s *t*-test for pairwise comparisons between the control and fructose groups. For analyses involving multiple groups, i.e., diet and fructose, two-way ANOVA followed by Tukey’s post hoc test was performed. Statistical significance is indicated by asterisks (*) when comparing fructose-supplemented groups to their respective controls: * *p* < 0.05; ** *p* < 0.01; *** *p* < 0.001; **** *p* < 0.0001; ns = not significant. Data are presented as mean ± SEM.

**Figure 2 nutrients-17-00124-f002:**
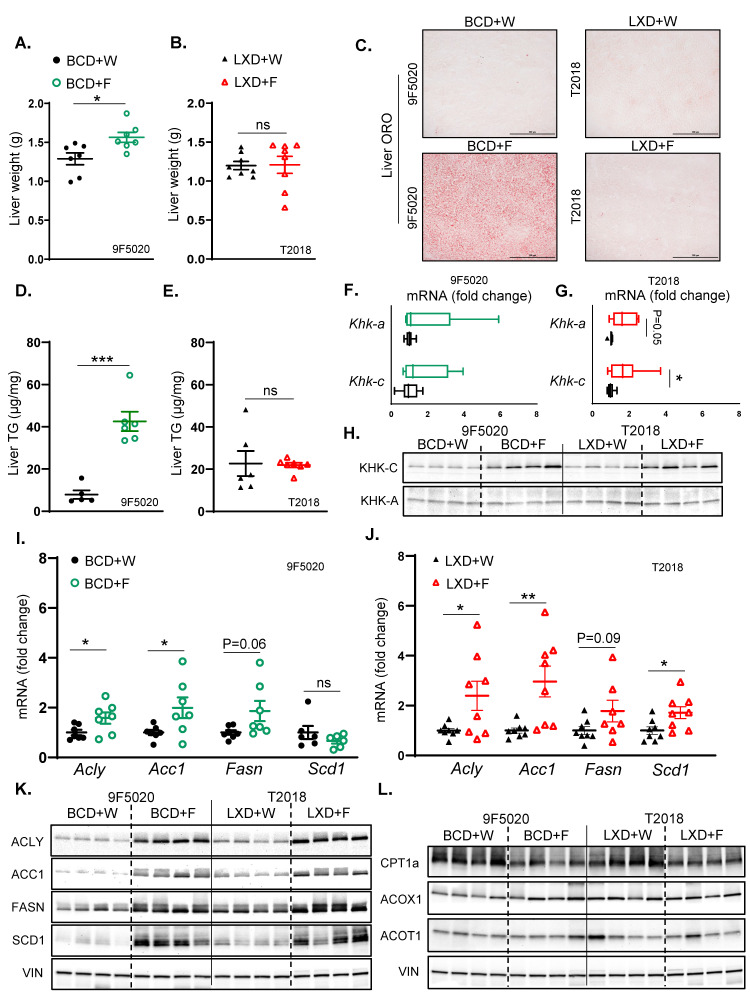
Fructose supplementation on BCD but not on LXD supports hepatic steatosis. (**A**) Liver weights of BCD-fed mice at sacrifice after 10 weeks on the diet. (**B**) Liver weights of LXD-fed mice at sacrifice after 10 weeks on the diet. (**C**) Oil Red O staining of liver sections from BCD and LXD-fed mice, showing hepatic lipid accumulation. (**D**) Hepatic triglyceride (TG) content in BCD-fed and (**E)** LXD-fed mice given fructose or regular water. (**F**) mRNA expression of KHK isoforms in the liver of BCD-fed (**G**) and LXD-fed mice. (**H**) Protein levels of KHK isoforms in the liver of BCD and LXD-fed mice, respectively. (**I**) Hepatic mRNA expression of enzymes involved in fatty acid synthesis in BCD-fed (**J**) and LXD-fed mice; *n* = 7–8 mice per group. (**K**) Protein levels of enzymes involved in hepatic fatty acid synthesis in the livers of BCD and LXD-fed mice, respectively. (**L**) Protein levels of enzymes involved in hepatic fatty acid oxidation in BCD and LXD-fed mice (*n* = 4 mice/group). Statistical comparisons were conducted using Student’s *t*-test for pairwise comparisons between the control and fructose groups. Statistical significance is indicated by asterisks (*) when comparing fructose-supplemented groups to their respective controls: * *p* < 0.05; ** *p* < 0.01; *** *p* < 0.001; ns = not significant. Data are presented as mean ± SEM.

**Figure 3 nutrients-17-00124-f003:**
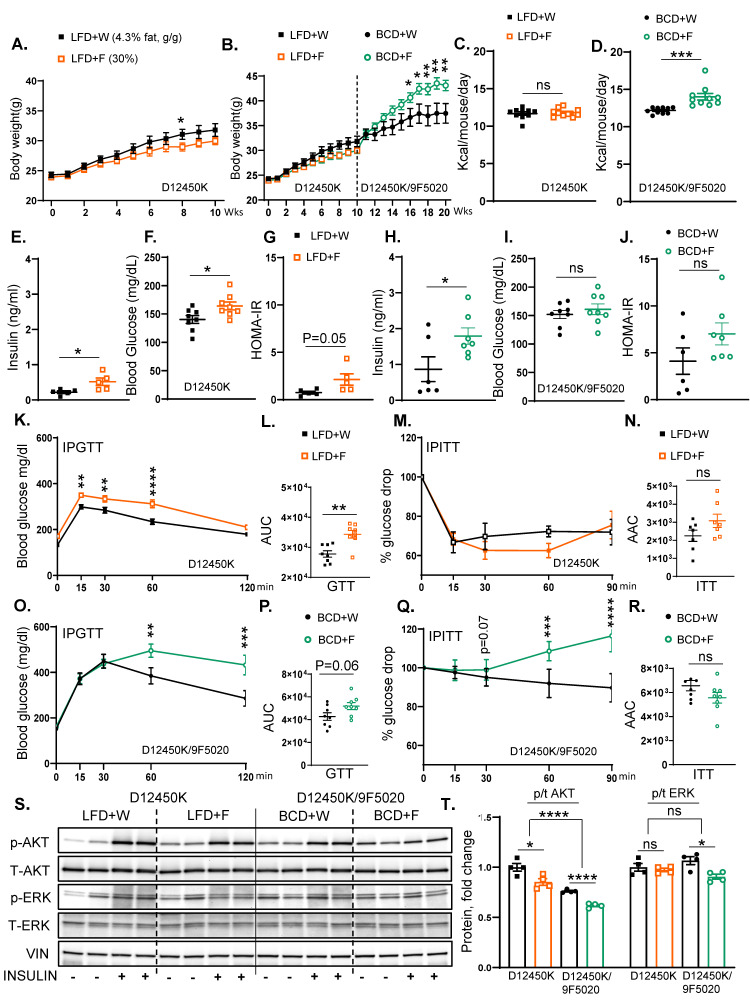
Switching from a low-fat diet to a Boston chow diet restores weight gain and the development of insulin resistance. (**A**) Weight gain of male mice on low-fat diet (D12450K) provided with either regular water (LFD + W) or 30% fructose-sweetened water (LFD + F) for 10 weeks, *n* = 8 mice per group. (**B**) Weight gain of male mice switched to the Boston chow diet, supplemented with either regular water (BCD + W) or 30% fructose-sweetened water (BCD + F) for an additional 10 weeks, *n* = 8 mice per group. (**C**) Total caloric intake in LFD-fed (**D**) and BCD-fed mice. (**E**) Fasting insulin levels, (**F**) blood glucose concentrations, and (**G**) HOMA-IR in LFD-fed mice measured after 10 weeks on the diet. (**H**) Fasting insulin levels, (**I**) blood glucose concentrations, and (**J**) HOMA-IR for BCD-fed mice at 20 weeks on the diet. (**K**) Glucose tolerance test (GTT) results in LFD-fed mice after 8 weeks on the diet. (**L**) AUC for GTT results in LFD-fed mice. (**M**) Insulin tolerance test (ITT) performed in LFD-fed mice at 8 weeks on the diet and (**N**) area above the curve (AAC) for ITT in LFD-fed mice. (**O**) GTT results for BCD-fed mice at 18 weeks on the diet and (**P**) AUC analysis for GTT results in BCD-fed mice. (**Q**) ITT results for BCD-fed mice at 18 weeks on the diet and (**R**) AAC analysis for ITT in BCD-fed mice. (**S**) Western blot (WB) analysis of phosphorylated and total AKT and ERK proteins in LFD- and BCD-fed mice, respectively, and (**T**) Image J quantification showing protein levels of phosphorylated and total AKT and ERK (*n* = 4 mice/group). Statistical comparisons were conducted using Student’s *t*-test for pairwise comparisons between the control and fructose groups. For analyses involving multiple groups, i.e., diet and fructose, two-way ANOVA followed by Tukey’s post hoc test was performed. Statistical significance is indicated by asterisks (*) when comparing fructose-supplemented groups to their respective controls: * *p* < 0.05; ** *p* < 0.01; *** *p* < 0.001; **** *p* < 0.0001; ns = not significant. Data are presented as mean ± SEM.

**Figure 4 nutrients-17-00124-f004:**
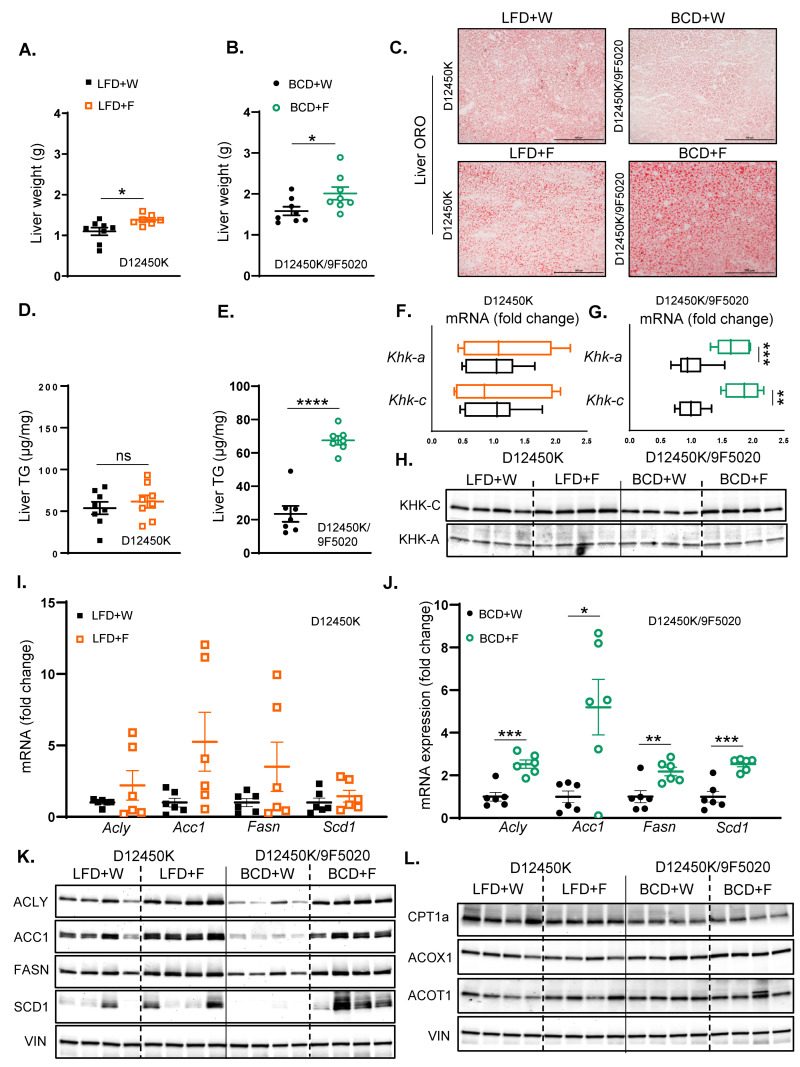
Fructose enhances hepatic steatosis more strongly in mice on BCD than on LFD. (**A**) Liver weights of LFD-fed mice at sacrifice after 10 weeks on the diet. (**B**) Liver weights of BCD-fed mice at sacrifice after 20 weeks on the diet. (**C**) Oil Red O staining of liver sections from LFD- and BCD-fed mice, highlighting hepatic lipid accumulation. (**D**) Hepatic triglyceride (TG) content in LFD-fed (**E**) and BCD-fed mice. (**F**) mRNA expression of KHK isoforms in the liver of LFD-fed (**G**) and BCD-fed mice. (**H**) Protein levels of KHK isoforms in the liver of LFD- and BCD-fed mice. (**I**) mRNA expression of enzymes involved in hepatic fatty acid synthesis in LFD-fed (**J)** and BCD-fed mice. *n* = 6 mice per group. (**K**) Protein levels of enzymes involved in hepatic fatty acid synthesis in the liver of LFD and BCD-fed mice. (**L**) Protein levels of enzymes involved in hepatic fatty acid oxidation in LFD and BCD-fed mice (*n* = 4 mice/group). Statistical comparisons were conducted using Student’s *t*-test for pairwise comparisons between the control and fructose groups. Statistical significance is indicated by asterisks (*) when comparing fructose-supplemented groups to their respective controls: * *p* < 0.05; ** *p* < 0.01; *** *p* < 0.001; **** *p* < 0.0001; ns = not significant. Data are presented as mean ± SEM.

**Figure 5 nutrients-17-00124-f005:**
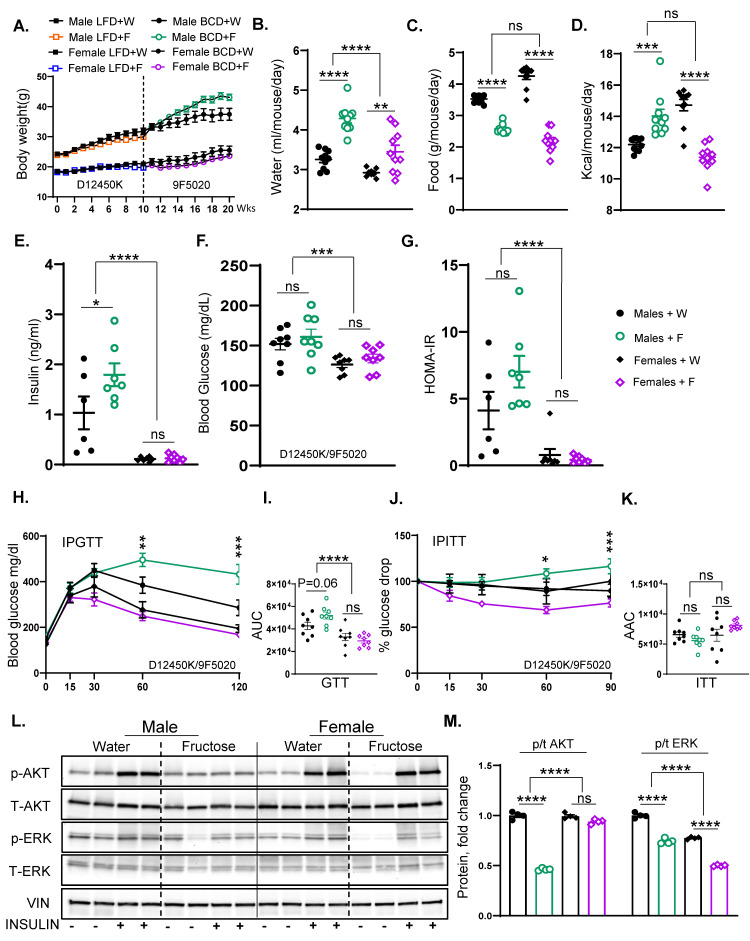
Fructose supplementation results in weight gain and insulin resistance in male but not female mice. (**A**) Weight gain in male and female mice on LFD for 10 weeks who were then switched over to BCD for another 10 weeks and supplemented with either regular water or 30% fructose-sweetened water (*n* = 16 mice/group for first 10 weeks and *n* = 8 mice/group for the next 10 weeks). (**B**) Water consumption, (**C**) food consumption, and **(D**) total calorie intake of male and female mice from weeks 11 to 20 on the diets. (**E**) Fasting insulin levels, (**F**) blood glucose concentrations, and (**G**) HOMA-IR in male and female mice after 20 weeks on the diet. (**H**) Glucose tolerance test (GTT) results (**I**) and area under the curve (AUC) analysis for male and female mice, performed after 18 weeks on the diet. (**J**) Insulin tolerance test (ITT) and (**K**) area above the curve (AAC) analysis for male and female mice, performed after 18 weeks on the diet. (**L**) Western blot analysis (WB) of phosphorylated and total AKT and ERK proteins in liver tissues of male and female mice, respectively, and (**M**) Image J quantification showing protein levels of phosphorylated and total AKT and ERK (*n* = 4 mice/group). Statistical analysis was performed using two-way ANOVA followed by Tukey’s post hoc test for comparisons across multiple groups. Statistical significance, denoted by asterisks (*), represents comparisons of fructose-supplemented groups to their respective controls, as follows: * *p* < 0.05; ** *p* < 0.01; *** *p* < 0.001; **** *p* < 0.0001; ns = not significant. All data are expressed as mean ± SEM.

**Figure 6 nutrients-17-00124-f006:**
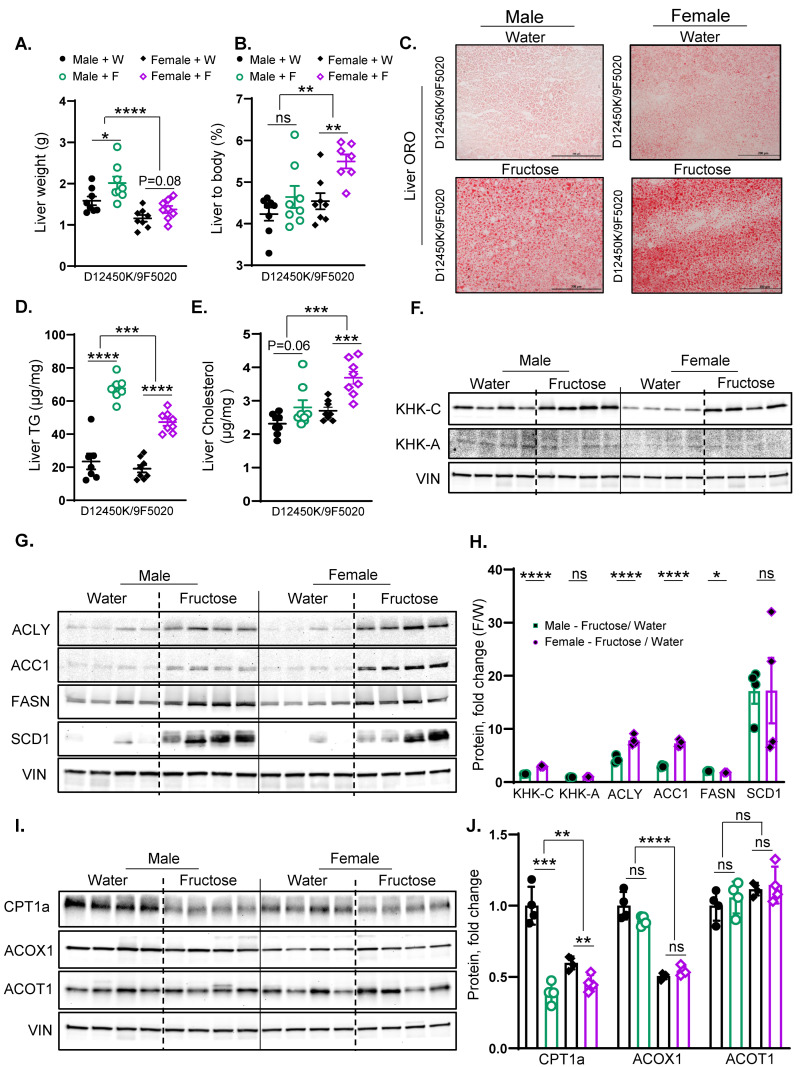
Fructose supplementation in female mice more strongly upregulates KHK-C and DNL enzymes than in males. (**A**) Liver weights of male and female mice at the time of sacrifice after 20 weeks on the diet. (**B**) Liver-to-body weight ratio in male and female mice. (**C**) Oil Red O staining of liver sections from male and female mice, showing lipid accumulation. (**D**) Hepatic triglyceride (TG) content (**E**) and liver cholesterol content in male and female mice after 20 weeks on the diet. (**F**) Protein levels of KHK isoforms in the liver of male and female mice. (**G**) Protein levels of enzymes involved in the hepatic fatty acid synthesis; (**H**) and ImageJ quantification of these proteins in the livers of male and female mice. (**I**) Protein levels of enzymes involved in hepatic fatty acid oxidation; (**J**) and ImageJ quantification of these proteins in male and female mice (*n* = 4 mice/group). Statistical analysis was performed using two-way ANOVA followed by Tukey’s post hoc test for comparisons across multiple groups. Statistical significance, denoted by asterisks (*), represents comparisons of fructose-supplemented groups to their respective controls, as follows: * *p* < 0.05; ** *p* < 0.01; *** *p* < 0.001; **** *p* < 0.0001; ns = not significant. All data are expressed as mean ± SEM.

## Data Availability

The original contributions presented in this study are included in the article/Appendix A. Further inquiries can be directed to the corresponding author.

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
