# Peer review of "Fructose-Induced Metabolic Dysfunction Is Dependent on the Baseline Diet, the Length of the Dietary Exposure, and Sex of the Mice"

_nutrients, 2024, doi:10.3390/nu17010124_

Round 1
Reviewer 1 Report
Comments and Suggestions for Authors
This manuscript is very relevant, because it explores a very controversial area in nutri-metabolic research. The manuscript is mostly clear, however, it needs some improvements. Long sentences can be confusing, please try to restructure into short sentences. This will increase the clarity of the manuscript. Please use consistent terminology throughout the manuscript.
Title: it would be more relevant to highlight the impact of the basal diet. For example “Impact of basal diet, duration of exposure and sex on fructose-induced metabolic dysfunction in mice”
In the abstract, please break up long sentences, and avoid repeating sex differences in 2 sections. It is sufficient in the results.
The introduction is complex, a bit too focused on the paradox of the effects of fructose on metabolic health. Try to avoid repetitive information.
Please reorganize the materials and methods in a clearer form, and include the description of technical details only if absolutely necessary.
The results are well structured, with clear results, legible figures. I have only one remark, please avoid repeating the results presented in both figures and text.
The discussions are complex, include the results. Regarding limitations, please take into account the microbiota as a perspective in the future.
The conclusions are clear and relevant.
After correcting them, the article can be published.
Author Response
We thank the editor and the reviewers for critically appraising our manuscript. The thoughtful suggestions helped us streamline the information presented and increase the clarity of our findings. Please, see our point-by-point response in blue presented below.
Reviewer comments:
Reviewer1
This manuscript is very relevant because it explores a very controversial area in nutri-metabolic research. The manuscript is mostly clear; however, it needs some improvements. Long sentences can be confusing, please try to restructure into short sentences. This will increase the clarity of the manuscript. Please use consistent terminology throughout the manuscript.
We thank the reviewer for constructive criticism and his/her suggestions on how to improve the manuscript. We have now increased the clarity of our descriptions and broke down complex sentences into easy to follow statements. The changes in the manuscript are highlighted in red.
Title: it would be more relevant to highlight the impact of the basal diet. For example, “Impact of basal diet, duration of exposure and sex on fructose-induced metabolic dysfunction in mice”
We agree with the reviewer and have now revisited the title as suggested. The new title is “Fructose-induced metabolic dysfunction is dependent on the baseline diet, but also on the length of the dietary exposure and sex of the mice.”
In the abstract, please break up long sentences, and avoid repeating sex differences in 2 sections. It is sufficient in the results.
We have now revised the abstract by breaking up the long sentences and included descriptions of sex differences only when needed. We have also made changes to the abstract as requested by reviewer 2.
The introduction is complex, and a bit too focused on the paradox of the effects of fructose on metabolic health. Try to avoid repetitive information.
Thank you for pointing this out. We have now expanded the introduction section by including more information about the obesity epidemic.
Please reorganize the materials and methods in a clearer form, and include the description of technical details only if absolutely necessary.
In Section 2 (Materials and Methods), we separated the H&E staining technique (2.3) from the ORO staining technique (2.4) to improve clarity and organization. Additionally, we expanded and provided a more detailed explanation of the glucose and insulin tolerance test technique (2.2) to enhance understanding. Furthermore, we revised the title of subsection 2.5 from "Assays Quantifying Serum and Liver Metabolic Dysfunction" to "Measuring Quantitative Hepatic Triglycerides" for more clarity. We have also deleted the unnecessary details, such as the order in which the mice were sacrificed.
The results are well structured, with clear results, and legible figures. I have only one remark, please avoid repeating the results presented in both figures and text.
We thank the reviewer for praising our results section. The figures visually summarize the data, while the text helps to interpret the key findings. Our intent was to provide a clear connection between the figures and the text. We have made our best attempt to reduce the overlapping information.
The discussions are complex, including the results. Regarding limitations, please take into account the microbiota as a perspective in the future.
We have now simplified our discussions. Moreover, we highlighted microbiota as a future research perspective in a new section titled: Future Perspectives and Directions, Section 7, page 26. In future research, we plan to incorporate microbiota studies to investigate it’s possible role in mediating fructose-induced metabolic dysfunction.
The conclusions are clear and relevant.
Thank you!
After correcting them, the article can be published.
We greatly appreciate the reviewer's enthusiasm for our study.
Reviewer 2 Report
Comments and Suggestions for Authors
Before the work conducted by Fadhul et al. can be considered for publication in Nutrients, I suggest the following revisions.
Specify your study’s objectives in the abstract. More details about methods should also be provided in this section.
An international and wider overview of the addressed topics in your research should be included in the Introduction. In its current state is not robust enough.
More details regarding the glucose and insulin tolerance test could be provided in subsection 2.2.
I encourage the authors to include a flowchart in section 2 with all the steps carried out in the investigations. This would help the readers to visualize the logical sequence of all the procedures.
The Results section has to be revised. There are several excerpts to be moved to other sections once here you should only report the obtained results and present tables and figures. The description of applied methodologies should be moved to section 2 and discussions should be moved to section 4.
Some parts of section 4 are redundant, like the first statement. Here, you should go directly to the discussion of your results and compare them with other relevant studies on the topic.
Line 704-743: So much text without any cited reference. This is not adequate for a Discussion section and must be revised.
The study’s strengths and limitations should be pointed out in a separate section after the Discussion
Conclusions have to be improved and more is expected to be stated in this section. Future perspectives and directions for further studies are crucial. Why your study is novel and impactful?
Author Response
Reviewer2
Before the work conducted by Fadhul et al. can be considered for publication in Nutrients, I suggest the following revisions.
Specify your study’s objectives in the abstract. More details about methods should also be provided in this section.
Thank you for pointing this out. The study’s objectives have been incorporated into the Background/Objectives section of the abstract. Furthermore, more details about the different normal chow diets and their dietary composition have been included in the Methods section of the abstract. Our revisions are highlighted in red for easier review.
An international and wider overview of the addressed topics in your research should be included in the Introduction. In its current state is not robust enough.
We have now incorporated more detailed descriptions providing an international perspective on the rise in obesity following the COVID-19 pandemic in the Introduction, Section 1, page 3.
More details regarding the glucose and insulin tolerance test could be provided in subsection 2.2.
We have now provided more detailed descriptions regarding GTT and ITT in Subsection 2.2 (Materials and Methods), page 6. We have also described AAC and AUC.
I encourage the authors to include a flowchart in section 2 with all the steps carried out in the investigations. This would help the readers to visualize the logical sequence of all the procedures.
Thank you for the suggestion. We have included detailed experimental designs for each experiment in the supplemental figures. These figures illustrate the timeline, including the start and end weeks of the experiments, as well as data for GTT, ITT, and weekly measurements of weight and food intake. These experimental designs are presented in Supplemental Figure 1B (page 1), Supplemental Figure 4A (page 5), and Supplemental Figure 6A (page 7).
The Results section has to be revised. There are several excerpts to be moved to other sections once here you should only report the obtained results and present tables and figures. The description of applied methodologies should be moved to section 2 and discussions should be moved to section 4.
Thank you for your feedback. We understand your concern, but we believe keeping some brief methodological details helps the reader better understand the findings. Moreover, making conclusions about a set of data is a good segway into a new set of data. When writing our papers we follow a style by Strunk & White, in The Elements of Style. This style of writing has helped us publish several manuscripts in high-impact journals. Please see examples: PMID: 36822479; PMID: 31577934; PMID: 37230214.
Some parts of section 4 are redundant, like the first statement. Here, you should go directly to the discussion of your results and compare them with other relevant studies on the topic.
We revised the first statement of the discussion section and modified the first paragraph as instructed by the author. Again some of these differences are driven by the style of scientific writing that we use.
Line 704-743: So much text without any cited reference. This is not adequate for a Discussion section and must be revised.
This section is a summary of our results so no references were provided. As stated above all our previous publications start this way. We hope that the reviewer appreciates that we follow a slightly different style of writing as documented above.
As a response to the reviewer's comment, we have added 6 references to our manuscript.
The study’s strengths and limitations should be pointed out in a separate section after the Discussion
Thank you for pointing this out. We have now included a separate detailed section for the limitations of the study in Section 5, of the discussion.
Conclusions have to be improved and more is expected to be stated in this section. Future perspectives and directions for further studies are crucial. Why your study is novel and impactful?
We have expanded Section 6 (Conclusion) to provide more details about the significance of this study as requested by the reviewer. Additionally, we have added a new Section 7 (Future Directions) on page 26, including plans to incorporate microbiota studies in future research.
Reviewer 3 Report
Comments and Suggestions for Authors
This is an interesting study. However, for the benefit of the reader, a number of points need clarifying and certain statements required further justification.
1. The content of Boston chow diet (BCD) is not much different from that of LCD. Only the fat content increases by 4% in BCD. Is it easier to induce the negative metabolic effects of fructose if the fat content exceeds 20% in the diet?
2. This study only revealed gender difference that may be an important factor leading to the occurrence of fructose-induced metabolic syndrome. Please cite some relevant literatures in the discussion section to explain whether humans have this tendency.
3. What is the difference between AUC and AAC? Can AAC be a biomarker for DM? Please also cite some relevant literature using AAC as a Biomarker.
4. The weights of adipose tissue should be described.
5. Please explain why fructose supplementation increased the calorie intake but decreased blood glucose, insulin and HOMA-IR in mice fed the LXD (Fig.1, H, I, J)
6. Please explain why fructose supplementation decreased the food intake but increased calorie intake in male mice fed the BCD (Fig.5, C, D).
7. About the original Western blot, there are some concerns as follows:
(1). Why these two proteins for p-AKT and p-ERK or T-AKT and T-ERK are expressed on same blot?
(2). Why are the molecular markers expressed on some blots, but not on others?
Author Response
Reviewer 3
This is an interesting study. However, for the benefit of the reader, a number of points need clarifying and
We thank the reviewer for their kind words and intellectual input to help us improve this manuscript. Our revisions based on the reveiwer’s comments are highlighted in red for easier review.
- The content of the Boston chow diet (BCD) is not much different from that of LCD. Only the fat content increases by 4% in BCD. Is it easier to induce the negative metabolic effects of fructose if the fat content exceeds 20% in the diet?
We believe the reviewer is absolutely correct. The following sentence was added to our discussion section “Our study was not designed to determine the minimal amount of dietary fat required to induce fructose derangements, but this appears to be greater than twenty percent.”
- This study only revealed gender differences that may be an important factor leading to the occurrence of fructose-induced metabolic syndrome. Please cite some relevant literature in the discussion section to explain whether humans have this tendency.
The reviewer is correct. There are substantial sex differences in terms of metabolic dysfunction. However, the difference in fructose metabolism between men and women is unknown. There is at least one study in humans indicating that there are differences between men and women in terms of their FGF21 response after fructose stimulation (PMID: 31687174). This agrees with the sex differences reported in our study. This reference has been added to the discussion section as requested.
- What is the difference between AUC and AAC? Can AAC be a biomarker for DM? Please also cite some relevant literature using AAC as a Biomarker.
We thank the reviewer for asking us to clarify these points. The AUC and AAC are measures of glucose exertion during GTT and ITT tests, respectively. During GTT, the subjects are given glucose and blood glucose levels increase over time. AUC captures that increase in glucose over baseline for the duration of the test. Similarly, during ITT, the subjects are given insulin and blood glucose levels decrease over time. AAC captures that decrease in glucose under baseline for the duration of the test.
As the reviewer suggested, AAC can be used as an index of glucose intolerance (PMID: 30603243). This discussion and reference have been added to methods section 2.2 titled “Glucose and insulin tolerance tests”
- The weights of adipose tissue should be described.
The mass of perigonadal adipose tissue is detailed in subsection 3.5 of the Results, on page 19 for our third experiment. Unfortunately, we did not collect adipose tissue from the first two experiments in mice on BCD or LXD.
- Please explain why fructose supplementation increased the calorie intake but decreased blood glucose, insulin, and HOMA-IR in mice fed the LXD (Fig.1, H, I, J)
The reviewer is bringing up an excellent point which is the foundation of our study. As mentioned in our introduction section some studies indeed document improvement in glucose and insulin response with fructose feeding PMID: 7002511. Therefore, early research on fructose in the 80s and 90s entertained the possibility that fructose may be used as an alternative sweetener for patients with diabetes PMID: 8116561. As the quality of our Western diet continued to deteriorate the effects of fructose on metabolic outcomes have largely turned out to be undesirable. This important point has been added as a new paragraph to the discussion section.
- Please explain why fructose supplementation decreased the food intake but increased calorie intake in male mice fed the BCD (Fig.5, C, D).
In sup figure 1C and 1D we show that mice on BCD kept the same solid food intake in spite of receiving additional calories from fructose water. That resulted in higher caloric intake. In Figure 5, the mice were first on LFD and they were accustomed to eating less solid diet when given fructose in water. When the mice were switched to BCD they were still accustomed to eating less solid food when given fructose but the decrease was diminished compared to the LFD (Sup Fig 4B-E). Thus, in Fig 5C, D, the mice consumed less solid BCD when given fructose. However, this decrease was not as large to offset the calories consumed from fructose, which resulted in higher caloric intake.
- About the original Western blot, there are some concerns as follows:
(1). Why these two proteins for p-AKT and p-ERK or T-AKT and T-ERK are expressed on the same blot?
We combined the rabbit monoclonal anti-pAkt/cat (4060) and the rabbit monoclonal anti-pERK/ cat (4370) in one tube. Also, we combined the rabbit monoclonal anti-total Akt/ cat (4691) and the rabbit monoclonal anti-total ERK/ cat (9102), in one tube. Thus, we're measuring these two proteins on the same blot.
Dr. Softic trained with C. Ronald Kahn, MD who discovered how the insulin receptor works and downstream insulin signaling. Dr. Kahn has many decades of experience combining Akt and ERK western blots on the same gel and dr. Softic continued that tradition to save the reagents and expedite the work.
(2). Why are the molecular markers expressed on some blots, but not on others?
We submitted uncut western blots to the journal three times. Initially, we submitted blot images that included molecular markers and labels. The journal asked us to remove them. Subsequently, we provided the same original blot images, but without molecular markers or labeling.
Round 2
Reviewer 2 Report
Comments and Suggestions for Authors
Some of my previous comments were not properly addressed:
The Results section has to be revised. There are several excerpts to be moved to other sections once here you should only report the obtained results and present tables and figures. The description of applied methodologies should be moved to section 2 and discussions should be moved to section 4.
Some parts of section 4 are redundant, like the first statement. Here, you should go directly to the discussion of your results and compare them with other relevant studies on the topic.
Line 704-743: So much text without any cited reference. This is not adequate for a Discussion section and must be revised.
Author Response
Reviewer 2
Comment 1: The Results section has to be revised. There are several excerpts to be moved to other sections once here you should only report the obtained results and present tables and figures. The description of applied methodologies should be moved to section 2 and discussions should be moved to section 4.
Response: Thank you for your feedback. We understand your concern, but we believe keeping some brief methodological details helps the reader better understand the findings. Moreover, making conclusions about a set of data is a good segway into a new set of data. When writing our papers we follow a style by Strunk & White, in The Elements of Style. This style of writing has helped us publish several manuscripts in high-impact journals. Please see examples: PMID: 36822479; PMID: 31577934; PMID: 37230214.
Comment 2: Some parts of section 4 are redundant, like the first statement. Here, you should go directly to the discussion of your results and compare them with other relevant studies on the topic.
Response: We revised the first statement of the discussion section and modified the first paragraph as instructed by the author. Again, some of these differences are driven by the style of scientific writing that we use.
Comment 3: Line 704-743: So much text without any cited reference. This is not adequate for a Discussion section and must be revised.
Response: This section is a summary of our results so no references were provided. As stated above all our previous publications start this way. We hope that the reviewer appreciates that we follow a slightly different style of writing as documented above.
Reviewer 3 Report
Comments and Suggestions for Authors
1. Generally, animals will have some discomfort with changes in feed type, but these discomforts will return to their original appetite within 2-3 days. Therefore, I cannot agree with the author’s reply to question 6.
2. Regarding to AAC, the literature (PMID: 30603243) only described AUC but didn’t study the AAC. Please cite some other literature to prove it.
3. In (PMID: 30603243) literature, “In clinical studies, fructose has either improved metabolic control of diabetic patients or caused no significant changes.”
But in animal studies, whether are there similar results? Please cite some literature to prove it.
4. I don't think a 4% difference in dietary lipids would have such a negative metabolic effect. Please cite some literature to prove it.
Author Response
Reviewer 3
Comment 1: Generally, animals will have some discomfort with changes in feed type, but these discomforts will return to their original appetite within 2-3 days. Therefore, I cannot agree with the author’s reply to question 6.
Response: The reviewer is correct. The differences in food intake upon switching to different diets are usually transitory and short-lived. We were referring to a difference in carbohydrate-to-fat ratio as the driver of these differences. Once the mice are used to eating high carbohydrate diet they prefer to maintain similar carbohydrate ration in the new diet.
Another possible explanation regarding food intake presented in Sup Figure 1C and 1D versus Fig 5C, D is the length of dietary exposure and the age of the mice. While both groups of mice were on BCD, food intake in the former was calculated over 1 to 10 weeks, while in the latter it was calculated over 11 to 20 weeks on the diet.
Comment 2: Regarding to AAC, the literature (PMID: 30603243) only described AUC but didn’t study the AAC. Please cite some other literature to prove it.
Response: We apologize for the confusion. The AAC is not used in clinical practice to assess insulin sensitivity. However, it is used in rodent research similar to AUC. Please see the reference (PMID: 38198372) which has now been added to the methods section of our manuscript.
Comment 3. In (PMID: 30603243) literature, “In clinical studies, fructose has either improved metabolic control of diabetic patients or caused no significant changes.”
But in animal studies, whether are there similar results? Please cite some literature to prove it.
Response: We thank the reviewer for requesting data from mouse studies. We now cite a paper in mice that showed increasing sugar intake from 5 to 30% had no impact on body weight gain (PMID: 30017356). We cite another paper showing mice that drink fructose had worse glucose tolerance than the mice that consumed fructose in a solid diet, which improved glucose tolerance (PMID: 31255519).
Comment 4. I don't think a 4% difference in dietary lipids would have such a negative metabolic effect. Please cite some literature to prove it.
Response: As stated in our previous response we did not study what percent of fat is required to observe the obesogenic effects of fructose. Therefore, we cannot make a stronger statement in response to the reviewer’s question than we already did. In our first response we stated, “Our study was not designed to determine the minimal amount of dietary fat required to induce fructose derangements, but this appears to be greater than twenty percent.”